# Measurement of Hepatic CYP3A4 and 2D6 Activity Using Radioiodine-Labeled *O*-Desmethylvenlafaxine

**DOI:** 10.3390/ijms231911458

**Published:** 2022-09-28

**Authors:** Asuka Mizutani, Masato Kobayashi, Riku Aibe, Yuka Muranaka, Kodai Nishi, Masanori Kitamura, Chie Suzuki, Ryuichi Nishii, Naoto Shikano, Yasuhiro Magata, Yasushi Ishida, Munetaka Kunishima, Keiichi Kawai

**Affiliations:** 1Faculty of Health Sciences, Institute of Medical, Pharmaceutical and Health Sciences, Kanazawa University, 5-11-80 Kodatsuno, Kanazawa 920-0942, Ishikawa, Japan; 2Division of Health Sciences, Graduate School of Medical Sciences, Kanazawa University, 5-11-80 Kodatsuno, Kanazawa 920-0942, Ishikawa, Japan; 3Department of Radioisotope Medicine, Atomic Bomb Disease Institute, Nagasaki University, 1-12-4 Sakamoto, Nagasaki 852-8523, Nagasaki, Japan; 4Faculty of Pharmaceutical Sciences, Matsuyama University, 4-2 Bunkyo-cho, Matsuyama 790-8578, Ehime, Japan; 5Preeminent Medical Photonics Education & Research Center, Hamamatsu University School of Medicine, 1-20-1 Handayama, Higashi, Hamamatsu 431-3192, Shizuoka, Japan; 6Department of Molecular Imaging and Theranostics, Institute for Quantum Medical Science, Quantum Life and Medical Science Directorate, National Institutes for Quantum and Radiological Science and Technology, 4-9-1 Anagawa, Inage, Chiba 263-8555, Chiba, Japan; 7Department of Radiological Sciences, Ibaraki Prefectural University of Health Sciences, 4669-2 Ami, Inashiki 300-0394, Ibaraki, Japan; 8Department of Psychiatry, Faculty of Medicine, University of Miyazaki, 5200 Kihara, Kiyotake, Miyazaki 889-1692, Miyazaki, Japan; 9Faculty of Pharmaceutical Sciences, Institute of Medical, Pharmaceutical and Health Sciences, Kanazawa University, Kakuma, Kanazawa 920-1192, Ishikawa, Japan; 10Biomedical Imaging Research Center, University of Fukui, 23-3 Matsuokashimoaizuki, Eiheiji 910-1193, Fukui, Japan

**Keywords:** *O*-desmethylvenlafaxine, whole-body imaging, CYP3A4, CYP2D6, individualized medicine

## Abstract

Drug metabolizing enzyme activity is affected by various factors such as drug–drug interactions, and a method to quantify drug metabolizing enzyme activity in real time is needed. In this study, we developed a novel radiopharmaceutical for quantitative imaging to estimate hepatic CYP3A4 and CYP2D6 activity. Iodine-123- and 125-labeled *O*-desmethylvenlafaxine (^123/125^I-ODV) was obtained with high labeling and purity, and its metabolism was found to strongly involve CYP3A4 and CYP2D6. SPECT imaging in normal mice showed that the administered ^123^I-ODV accumulated early in the liver and was excreted into the gallbladder, as evaluated by time activity curves. In its biological distribution, ^125^I-ODV administered to mice accumulated early in the liver, and only the metabolite of ^125^I-ODV was quickly excreted into the bile. In CYP3A4- and CYP2D6-inhibited model mice, the accumulation in bile decreased more than in normal mice, indicating inhibition of metabolite production. These results indicated that imaging and quantifying the accumulation of radioactive metabolites in excretory organs will aid in determining the dosages of various drugs metabolized by CYP3A4 and CYP2D6 for individualized medicine. Thus, ^123/125^I-ODV has the potential to direct, comprehensive detection and measurement of hepatic CYP3A4 and CYP2D6 activity by a simple and less invasive approach.

## 1. Introduction

Drugs transit the body by four processes, absorption, distribution, metabolism, and elimination or excretion. Among these, metabolism is the process of transforming a drug within the body to make it more hydrophilic so that it can be excreted from the body [1,2,3]. In drug metabolism, drug metabolizing enzymes including cytochrome P450 (CYP) play an essential role, and their activity differs between individuals [4], so the clinical response to drug administration varies widely [5]. Individual specific responses to medication can be attributed to many complex factors. These factors include the genotypes of drug metabolizing enzymes; physiological conditions (age, sex, body size, and ethnicity); environmental influences (exposure to toxins, diet, and smoking); and pathological factors (liver and renal function, diabetes, obesity, and drug interactions), which can work alone or in combination to influence drug responses [6]. When estimating an individual’s drug responses, measurement of the activity and capacity of drug metabolizing enzymes is valuable for selecting the optimal drug and determining the prescribed dose for individualized medicine. It is ideal to measure the individual drug response in real-time. Although genotype analysis of CYP has been established, it only evaluates genetic polymorphisms and is not a comprehensive analysis. There have been attempts to evaluate CYP activity with various approaches such as blood or urine serial sampling [7,8]. However, special chemical analysis systems, including liquid chromatography and mass spectrometry, are not generally available in clinics or hospitals. We previously reported that whole-body imaging with ^123^I-iomazenil can directly quantify the activity of hepatic carboxylesterase by measuring the radioactive metabolite that accumulates in the gallbladder and urinary bladder [9]. To select radiopharmaceuticals to quantify the activity of hepatic drug-metabolizing enzymes, important conditions are as follows: (1) the radiopharmaceutical accumulates in a metabolic organ, (2) the radiopharmaceutical produces radioactive metabolites via specific drug-metabolizing enzymes, (3) only the radioactive metabolite is transferred from the metabolic organ to an excretory organ, and (4) the amount of accumulation in the excretory organ can be visualized and quantified. In this study, we developed a novel radiopharmaceutical for quantitative imaging to estimate hepatic CYP3A4 and CYP2D6 activity. Among the several CYP isozymes, CYP3A4 contributes to the metabolism of approximately 45% of all medical pharmaceuticals [10], and many prescribed drugs such as antidepressants, antipsychotics, and antiarrhythmics are metabolized mainly by CYP3A4. In addition, both CYP3A4 and CYP2D6 are involved in the metabolism of approximately 11% of all medical pharmaceuticals.

## 2. Results

### 2.1. Labeling of ^125^I-O-Desmethylvenlafaxine

Figure 1 shows an HPLC chromatogram of iodine-125-labeled *O*-desmethylvenlafaxine (^125^I-ODV) obtained by high-performance liquid chromatography (HPLC) with the γ-ray detector (reaction time, 5 min; chloramine-T, 2.0 × 10^−8^ mol). We confirmed peak 1 (retention time, 6.5–7.5 min), peak 2 (retention time, 8.5–10 min), and peak 3 (retention time, 12.5–13.5 min). To discriminate ^125^I-ODV, the reaction time and the amount of chloramine-T were changed. Figure 2 shows the percentage of peaks 1–3 by HPLC. The percentage of peak 3 increased while peak 1 decreased depending on the reaction time. The increased amount of chloramine-T resulted in a similar reaction.

The increased in reaction time and chloramine-T concentration can be a factor in the conformational change of ODV, peak 3 was estimated to be a compound in which the structure of ^125^I-ODV was changed by chloramine-T. Therefore, the peaks were identified as peak 2 being ^125^I-ODV, peak 3 being the byproduct of ^125^I-ODV, and peak 1 being ^125^I-NaI. The best condition of labeling was a reaction time of 5 min and chloramine-T concentration of 2.0 × 10^−8^ mol. After fractionation of ^125^I-ODV, acetonitrile was removed by nitrogen refluxing and adjusted to pH 7 with sodium hydroxide. The retention time of purified ^125^I-ODV was 8.5–10 min, that of ODV (no iodinated ODV) was 4.0–4.5 min, indicating that ^125^I-ODV is separated from ODV. The labeling index of ^125^I-ODV was 84.5 ± 3.6%. The radiochemical purity was more than 98% up to seven days at 4 °C.

### 2.2. Metabolism of ^125^I-ODV In Vitro

In this study, ^125^I-ODV and its metabolite were separated by HPLC (Figure 3). With an NADPH-generating system (NADPH [+]), an unknown radioactive metabolite (peak 4, retention time 4–5 min) and ^125^I-ODV (peak 5, retention time 17–18 min) were found, whereas without the NADPH-generating system (NADPH [–]), the unknown radioactive metabolite was not found. These results indicated that the fraction of the unknown radioactive metabolite was the NADPH-mediated metabolite of ^125^I-ODV. With the presence of NADPH [+], the peak of ^125^I-ODV metabolite significantly increased from about 0% to 46%, and the fraction of ^125^I-ODV significantly decreased from about 85% to 54% (Figure 4).

Figure 5 shows the metabolite rates of ^125^I-ODV. In the absence of NADPH [−] and with paroxetine and ketoconazole, the percentages of metabolite were 1.52%, 9.99%, and 17.6%, respectively, and the metabolite of ^125^I-ODV was significantly decreased compared with NADPH [+].

### 2.3. Whole-Body Imaging of ^123^I-ODV and Metabolite of ^123^I-ODV in Normal Mice

Figure 6 shows SPECT images in normal mice. Accumulation is shown in the lung, liver, and urinary bladder (frame 1). Though the accumulation in the lung and liver gradually decreased, in the urinary bladder it increased (frame 3). In addition, the accumulation moved to the gallbladder and intestines, and the contrast between the liver and gallbladder is clear (frame 6).

Figure 7 shows VOI analysis in normal mice of the liver and gallbladder. The accumulation in the liver decreased, while that in the gallbladder was gradually increased.

### 2.4. Metabolism of ^125^I-ODV in Bile of Normal and CYP-Inhibited Mice

In the in vitro thin layer chromatography (TLC) study, Rf values of ^125^I-NaI and ^125^I-ODV in bile were in ranges of 0.75–0.85 and 0.40–0.50, respectively. In the bile in normal and CYP-inhibited mice injected with ^125^I-ODV, the fractions of ^125^I-NaI and ^125^I-ODV were not found, and the fraction of ^125^I-ODV metabolite (Rf value; 0.10–0.30) was confirmed (Figure 8). The fractional ratio of the metabolite was more than 99%.

### 2.5. Biological Distribution of ^125^I-ODV and Metabolite of ^125^I-ODV in Normal and CYP-Inhibited Mice

Table 1 shows the accumulation of ^125^I-ODV and metabolite of ^125^I-ODV in normal mice. ^125^I-ODV was rapidly distributed to organs in the whole body. The accumulation in the lungs and kidneys significantly increased to more than 20% after 5 min of administration, and then it gradually decreased. The accumulation in the liver immediately increased after injection and then gradually decreased, whereas the accumulation in the gallbladder gradually increased after injection. In the thyroid and stomach, the accumulation was low at all time points.

Table 2 shows the accumulation of ^125^I-ODV and metabolite of ^125^I-ODV in CYP3A4-inhibited mice. The accumulation in the liver was comparable to the control at all time points, and the accumulation in the gallbladder was significantly lower than in the control 30 min later.

Table 3 shows the accumulation of ^125^I-ODV and metabolite of ^125^I-ODV in CYP2D6-inhibited mice. The accumulation in the liver was comparable to the control at all time points. In the gallbladder, the accumulation at 5 min was significantly lower than in the control, while the accumulation at 15 min was significantly higher than in the control.

## 3. Discussion

In this study, established to quantify hepatic CYP3A4 and CYP2D6 activity were quantified using newly labeled ^123/125^I-ODV. 

In general, non-metabolized radiopharmaceuticals are ideal in nuclear medicine, because radiopharmaceuticals target specific functional proteins in vivo. In our previous study, ^123^I-iomazenil was metabolized by hepatic carboxylesterase, and then almost all the radioactive metabolite moved into the gallbladder and urinary bladder [10]. We can measure hepatic carboxylesterase activity by detecting and quantifying the accumulation of a radioactive metabolite in the gallbladder and/or urinary bladder. Using the same technique, here we quantified hepatic CYP activity. It is often difficult to prescribe drugs for psychiatric pharmacotherapy in the clinic because the therapeutic index differs substantially between individuals. CYP2D6 as well as CYP3A4 contribute to the clearance of many clinical drugs, and many drugs are metabolized by both, including antidepressants, antipsychotics, and analgesics. Therefore, measurement of CYP3A4 and CYP2D6 activity by a less invasive and simple approach is important for the prescription of medication. ODV is a candidate mother compound and substrate of CYP3A4 as a new radiopharmaceutical. In the metabolic pathway of ODV, it is primarily converted to *N*, *O*-didesmethylvenlafaxine. Hydroxylation is the main metabolic reaction of ODV, and it mainly involves CYP3A4 [11]. Although there are no reports of ODV being metabolized by CYP2D6, we expected CYP2D6 to be contributing to the metabolism of radioiodine-labeled ODV.

We were able to establish a stable radio-labeling method for ODV, and ^125^I-ODV showed very high labeling, yield, and purity (Figure 2). In NADPH-mediated metabolism of ^125^I-ODV in vitro, to confirm the involvement of CYP, the effect of an NADPH-generating system was examined. With NADPH [+], ^125^I-ODV and an unknown radioactive metabolite of ^125^I-ODV were detected. In contrast, with NADPH [−], only ^125^I-ODV was detected (Figure 3). Thus, an unknown radioactive metabolite was produced when the NADPH-generating system provided energy for CYP, which was a metabolite of ^125^I-ODV. To further study the metabolism of ^125^I-ODV in vitro and to clarify the specific isozyme of CYP responsible for the metabolite of ^125^I-ODV, the effects of inhibitors on the metabolism were examined (Figure 5). Paroxetine and ketoconazole significantly inhibited the metabolism of ^125^I-ODV (79% and 63% inhibition). Therefore, ^125^I-ODV was mainly metabolized by CYP3A4 and CYP2D6. Although non-labeled ODV is metabolized only by CYP3A4 [11], ^125^I-ODV was found to be metabolized by both CYP3A4 and CYP2D6.

On imaging after ^123^I-ODV injection, the accumulation in the liver decreased, while accumulation in the gallbladder increased over time (Figure 7). To set a VOI on SPECT images, it is necessary that the accumulation in the targeted organ does not overlap and is clear. In normal mice, the gallbladder/liver ratio increased over time (0.62 to 11.43, Figure 7). Thus, ^123^I-ODV was excreted into the bile, confirming that the VOI can be easily set in the gallbladder. To confirm the radioactive materials in bile, gallbladders were collected from ^125^I-ODV-injected mice 60 min after injection. As shown in Figure 8, metabolites of ^125^I-ODV that are not ^125^I^−^ or ^125^I-ODV were found to be present in bile at a rate of more than 99%. This result indicated that ^125^I-ODV in the mouse liver was metabolized, and the metabolite of ^125^I-ODV was selectively excreted to the bile. Similarly, in CYP3A4- and CYP2D6-inhibited mice, only ^125^I-ODV was found in the bile. Thus, we found that inhibition of ^125^I-ODV metabolism did not result in metabolites being produced and excreted by another metabolic pathway.

Imaging showed that the metabolite of ^123^I-ODV was excreted in the bile and VOI was more likely to be set in the gallbladder, so biological distribution was performed for a more quantitative evaluation. The biological distribution in normal mice based on examining each organ showed similar results to whole-body imaging. For the biological distribution of ^125^I-ODV and metabolite of ^125^I-ODV (Table 1), the accumulation of ^125^I-ODV and metabolite of ^125^I-ODV was high in the blood, lung, and liver (a metabolic organ), in addition to the gallbladder and kidney (excretory organs), immediately after injection. The accumulation in the liver decreased over time. By contrast, accumulation in the gallbladder and kidney increased early on. These results indicate that ^125^I-ODV is excreted via enterohepatic and renal routes. In addition, low accumulation in the thyroid and stomach indicated that ^125^I-ODV was almost never metabolized to ^125^I^−^. Thus, ^125^I-ODV accumulated and metabolized in the liver, and the metabolite of ^125^I-ODV was excreted to the gallbladder. Since in the in vitro study, ^125^I-ODV was mainly metabolized by CYP3A4 and CYP2D6, we focused on CYP3A4 and CYP2D6 in the in vivo study. In model mice with reduced CYP3A4 and CYP2D6 activity (Table 2 and Table 3), the accumulation in the liver was similar to that in control mice at all time points. In contrast, in the model mice with reduced CYP3A4, the accumulation in the gallbladder at 5 and 15 min after injection of ^125^I-ODV was not significantly different from that of the control. However, at 30 and 60 min, the accumulation in the gallbladder was lower than that of control, indicating that the decrease in metabolite production due to reduced CYP3A4 activity was confirmed as a decrease in the amount of metabolites accumulation in the bile. In the model mice with reduced CYP2D6, the accumulation in the gallbladder was significantly lower at 5 min, higher at 15 min, and similar after 30 and 60 min compared to controls. The effect of paroxetine on CYP2D6 inhibition differed in each mouse and may have a significant effect on the variability of the accumulation in the gallbladder of the metabolite of ^125^I-ODV. Therefore, it is difficult to create a mouse model using CYP inhibitors to evaluate CYP activity, but this could be improved by using CYP-deficient mice.

These results indicate that ^123/125^I-ODV meets four important conditions for quantitative imaging of CYP activity: (1) ^123/125^I-ODV accumulates in the liver (a metabolic organ), (2) ^123/125^I-ODV is metabolized specifically by CYP3A4 and CYP2D6, and the metabolite is radioactive, (3) the metabolite of ^123/125^I-ODV transfers from the liver to the gallbladder, and (4) accumulation in the excretory organs can be visualized and quantitated.

## 4. Materials and Method

### 4.1. Materials

Methanol, chloramine-T, ethylene diamide tetra acetic acid (EDTA), and glucose-6-phosphate (G6P) were purchased from Nacalai tesque (Kyoto, Japan). β-Nicotinamide-adenine dinucleotide phosphate (β-NADP^+^), sodium hydroxide and isoflurane were purchased from Fujifilm Wako Pure Chemical Industries (Osaka, Japan), and glucose-6-phosphate dehydrogenase (G6PD) was purchased from Oriental Yeast (Osaka, Japan). *O*-Desmethylvenlafaxine was purchased Tokyo Chemical Industry (Tokyo, Japan). ^125^I-NaI was purchased from American Radiolabeled Chemicals (St. Louis, MO, USA). ^123^I-NaI was purchased from Fujifilm Toyama Chemical (Tokyo, Japan). All other reagents and chemicals were of analytical or high-performance liquid chromatography HPLC grade.

### 4.2. Labeling of ^125^I-ODV

ODV is a pharmacologically active metabolite of venlafaxine, which is a clinical antidepressant. ODV is mainly metabolized to *N*, *O*-didesmethylvenlafaxine by CYP3A4 [11]. Thus, ODV was selected as a substrate of CYP3A4 as a parent compound for a novel radiopharmaceutical with the expectation that it would be a substrate for CYP3A4.

ODV was labeled by radioiodine (Figure 9). A total of 2.0 × 10^−9^ mol ODV was dissolved by 100 µL methanol, and 2.0 × 10^−9^, 2.0 × 10^−8^, and 2.0 × 10^−7^ mol chloramine-T was dissolved by 20 µL distilled water. These solutions were mixed with 3.7 MBq ^125^I-NaI and incubated at room temperature for 5–15 min. The separation and purification of ^125^I-ODV were performed by an HPLC system consisting of a pump (model L-7100, Hitachi, Japan), a UV detector (Chromaster 5410, Hitachi, Japan), and a γ-ray detector (model RLC-701, Hitachi-Aloka Medical, Japan) equipped with a 5C_18_ AR-II-column (4.6 × 250 mm; 5 µm, Nacalai tesque). A PowerChrom (ver. 2.3.3, eDAQ, Japan) was used for data processing. The mobile phase consisted of 70% 20 mM KH_2_PO_4_ buffer (pH 4.0) and 30% acetonitrile at flow rate of 1.0 mL/min. The elution of ^125^I-ODV was monitored at 254 nm UV. The stability of ^125^I-ODV was analyzed by TLC. The elution of ^125^I-ODVwas spotted onto a silica gel TLC plate 60 F_254_ (Merck, Darmstadt, Germany). The TLC plate and the TLC spots were developed using methanol/acetate at a ratio of 100:1. After development and complete drying, the TLC plates were cut into 21 fractions, and the radiography associated with each fraction was quantified using a γ-ray counter (AccuFLEXγ 7000, Aloka, Tokyo, Japan). The fractional ratios of ^125^I^−^ and ^125^I-ODV were calculated by dividing the radioactive counts for each fraction by the total radioactivity count.

### 4.3. Metabolism of ^125^I-ODV In Vitro

Animal studies were approved by the Animal Care Committee at Kanazawa University (AP-173851) and were conducted in accordance with international standards for animal welfare and institutional guidelines. For preparation of the pooled mouse liver homogenates (MLH), three six-week-old ddY mice (Japan SLC, Tokyo, Japan) were euthanized under anesthesia with isoflurane, and the removed livers were weighed. After adding Krebs–Ringer phosphate buffer (pH 7.4), the pooled livers were homogenized with an ultrasonic homogenizer (SONIFIER250, Branson, MO, USA), and the protein content was measured according to the bicinchoninic acid method [12]. The pooled MLH was stored at −80 °C.

For analysis of NADPH-mediated metabolism of ^125^I-ODV, NADPH was used as an energy source of CYP-mediated metabolism. Adding an NADPH-generating system produced a state in which CYP was active. NADPH-mediated metabolism of ^125^I-ODV was examined in a mixture consisting of an NADPH-generating system (0.5 mM β-NADP^+^, 5 mM MgCl_2_, 5 mM G6P, and 1 U/mL G6PD), 100 mM sodium potassium phosphate buffer (pH 7.4), 50 μM EDTA disodium salt, and 1000 μg protein/20 μL pooled MLH in a final volume of 250 μL (NADPH [+]). A mixture without the NADPH-generating system was used as the control (NADPH [−]) to confirm NADPH dependence. The samples were incubated at 37 °C for 5, 15, 30, and 60 min with gentle shaking, ethanol was added to stop the reactions, and the samples were centrifuged at 18,000 g for 5 min. The supernatants in each sample were analyzed by HPLC. The mobile phase consisted of 70% 20 mM KH_2_PO_4_ buffer (pH 4.0) and 30% acetonitrile at a flow rate of 1.0 mL/min (*n* = 3).

For inhibition analysis of specific isozymes of CYP-mediated metabolism of ^125^I-ODV, selective inhibitors of CYP isozymes employed in this study included α-naphthoflavone (an inhibitor of CYP1A1 and 1A2) [13], sulfaphenazole (CYP2C9) [13,14], fluconazole (CYP2C19) [15], paroxetine (CYP2D6) [13,16,17,18,19], 4-methylpyrazole (CYP2E1) [14], and ketoconazole (CYP3A4) [13,14,19]. These selective inhibitors were used to elucidate the CYP responsible for the biotransformation of ^125^I-ODV. Each inhibitor was dissolved at 10 µM (dimethyl sulfoxide 0.8% *v*/*v* final concentration). The inhibition of metabolism of ^125^I-ODV was examined in a mixture consisting of 50 µL of the NADPH-generating system (NADPH [+]), 100 mM sodium potassium phosphate buffer (pH 7.4), 50 µM EDTA disodium salt, 1000 µg protein/20 µL pooled MLH, and 25 µL inhibitor in a final volume of 250 µL. The sample was incubated at 37 °C for 15 min with gentle shaking. The reaction was stopped by adding 100 µL ethanol, and the sample was centrifuged for 5 min at 18,000 g. The final supernatant was analyzed by HPLC using 65% 20 mM KH_2_PO_4_ buffer (pH 4.0) and 35% acetonitrile at a flow rate of 1.0 mL/min (*n* = 3).

### 4.4. Whole-Body Imaging of ^123^I-ODV in Normal Mice

^123^I-ODV was labeled and purified by the same method as ^125^I-ODV. For whole-body SPECT imaging in normal mice, all acquisitions were performed using a U-SPECT-II/CT system (MILabs, Utrecht, The Netherlands). Three normal mice were injected with 14.5 MBq of ^123^I-ODV via the tail vein and whole-body SPECT images were acquired at 10 min/frame immediately after injection until 60 min under 1.5% isoflurane anesthesia. The images were reconstructed using the filter-backed projection method with 16 subsets and 6 iterations. The voxel size was set to 0.8 mm × 0.8 mm × 0.8 mm. Neither attenuation nor scatter correction was performed. Post-reconstruction smoothing filtering was applied using a Gaussian smooth 3D filter of 1.0 mm. Image display was performed using PMOD (ver. 3.7). On SPECT images, the volume of interest (VOI) was drawn around the liver and gallbladder, and each percent injected dose (%ID) was calculated over time.

### 4.5. Metabolism of ^125^I-ODV in Bile of Normal and CYP-Inhibited Mice

^125^I-ODV was prepared to a radioactive concentration of 30 MBq/mL by the addition of saline. For metabolite analysis in CYP-inhibited mice, we produced model mice with reduced CYP3A4 or CYP2D6 activity by selective inhibition using ketoconazole or paroxetine. CYP3A4-inhibited mice were injected intraperitoneally with 50 mg/kg of ketoconazole [20] at 60 min before injection of ^125^I-ODV, and CYP2D6-inhibited mice were injected intraperitoneally with 30 mg/kg of paroxetine [21] at 60 min before injection of ^125^I-ODV. Each of three mice was administered ^125^I-ODV via the tail vein (3.0 MBq/100 µL/mouse). At 30 min, the three fasted mice were euthanized under isoflurane, and each gallbladder was collected. The radioactive products in bile were analyzed by TLC using methanol/acetate at a ratio of 100:1. Bile was directly spotted onto a TLC plate.

### 4.6. Biological Distribution of ^125^I-ODV in Normal and CYP-Inhibited Mice

For biological distribution in normal mice, ^125^I-ODV was prepared to a radioactivity concentration of 185 kBq/mL by adding saline. A total of 20 mice fasted for 6 hr were administered ^125^I-ODV via the tail vein (18.5 kBq/100 µL/mouse). After 5, 15, 30, and 60 min, normal mice were euthanized under isoflurane anesthesia (*n* = 4), and the following tissues were collected: blood, brain, heart, lung, thyroid, gallbladder, liver, spleen, pancreas, stomach, small intestines, large intestines, and kidney. Tissues were weighed, and radioactivity was quantified using a γ-ray counter to calculate the %ID or percent injected dose per gram of tissue (%ID/g).

To examine the biological distribution in CYP-inhibited mice, we produced model mice in the same way as for the analysis of metabolism in bile. At 5, 15, 30, and 60 min after ^125^I-ODV injection, mice were euthanized under isoflurane (*n* = 4), and the same tissues were collected and quantified.

### 4.7. Statistical Analysis

All results represent the average of least three experiments and are expressed as the mean ± standard deviation. Data were analyzed using the F-test and Student’s *t* test, and *p* < 0.01 or 0.05 was considered statistically significant.

## 5. Conclusions

^123/125^I-ODV has the potential to direct, comprehensive detection and measurement of hepatic CYP3A4 and CYP2D6 activity by a simple and less invasive approach. Imaging and quantifying the accumulation of radioactive metabolites in excretory organs will aid in determining the dosages of various drugs metabolized by CYP3A4 and CYP2D6 for individualized medicine.

## Figures and Tables

**Figure 1 ijms-23-11458-f001:**
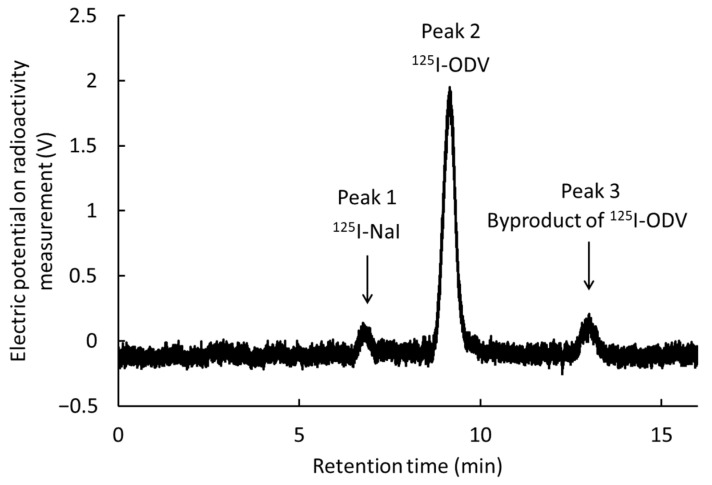
HPLC analysis of ^125^I-ODV. The chromatogram shows peaks 1−3.

**Figure 2 ijms-23-11458-f002:**
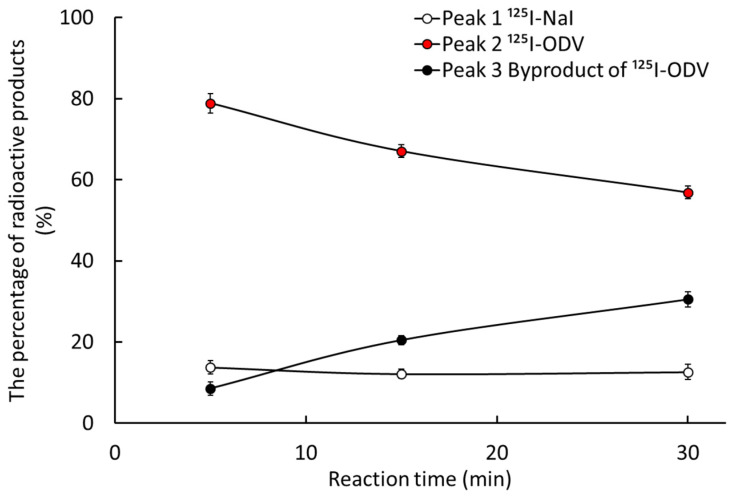
The percentages of peak areas from the HPLC chromatogram. Peak 1 was ^125^I-NaI. The percentage of peak 2 decreased and the percentage of peak 3 increased depending on the reaction time, so peak 2 was ^125^I-ODV and peak 3 was a byproduct of ^125^I-ODV.

**Figure 3 ijms-23-11458-f003:**
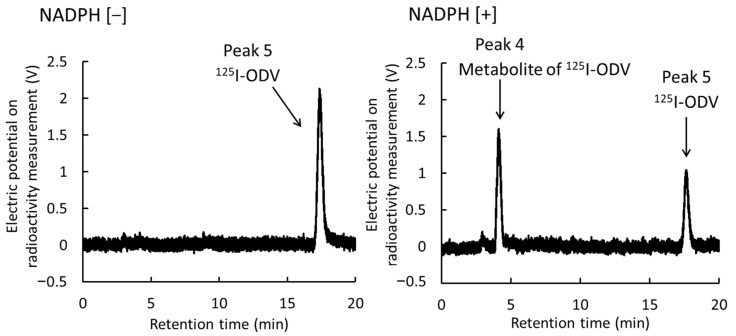
HPLC analysis of reaction mixtures (NADPH [−]/[+]). Peak 4 was not found for NADPH [−], so it was the metabolite of ^125^I-ODV dependent on the presence of NADPH. Peak 5 was ^125^I-ODV.

**Figure 4 ijms-23-11458-f004:**
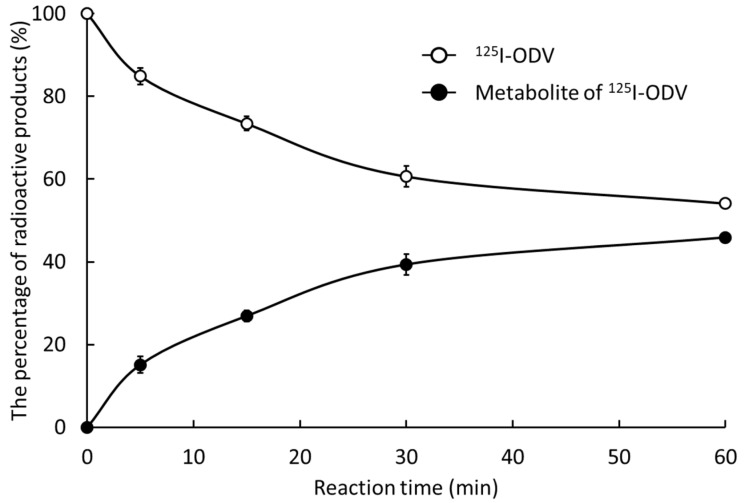
The percentage of ^125^I-ODV and its metabolite. Depending on the reaction time, ^125^I-ODV decreased and the metabolite of ^125^I-ODV increased. After 30 min of reaction time, the percentage of ^125^I-ODV and its metabolite became mostly constant.

**Figure 5 ijms-23-11458-f005:**
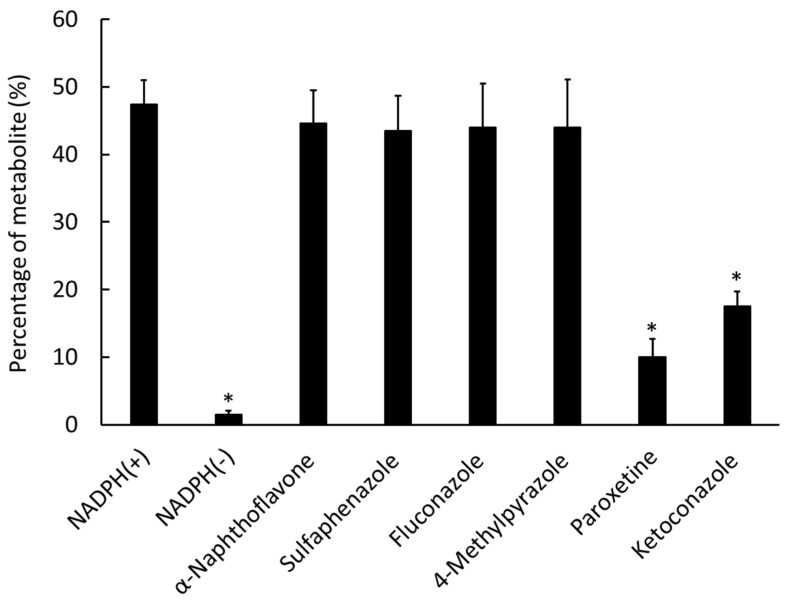
Effect of CYP inhibitors on the metabolism of ^125^I-ODV. The percentage of the metabolite was determined using mouse liver microsomes. With NADPH [−], paroxetine, and ketoconazole, the metabolite of ^125^I-ODV significantly decreased compared with NADPH [+]. * *p* < 0.01 compared with NADPH[+].

**Figure 6 ijms-23-11458-f006:**
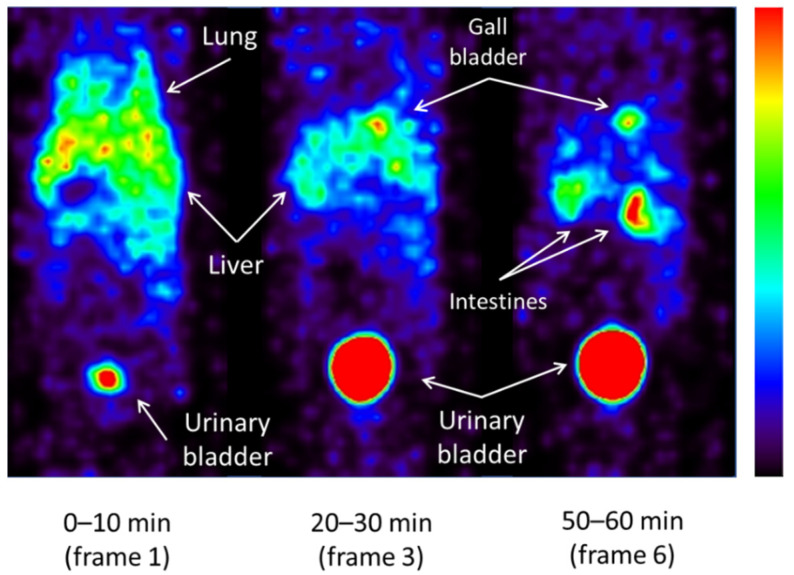
Whole-body images of a normal mouse obtained at 0–10, 20–30, and 50–60 min after 14.5 MBq ^123^I-ODV injection under 1.5% isoflurane anesthesia. Depending on the time, the accumulation in lung and liver decreased, and the accumulation in the gallbladder became clear. In addition, increased accumulation in the intestines and urinary bladder indicated that excretion was progressing.

**Figure 7 ijms-23-11458-f007:**
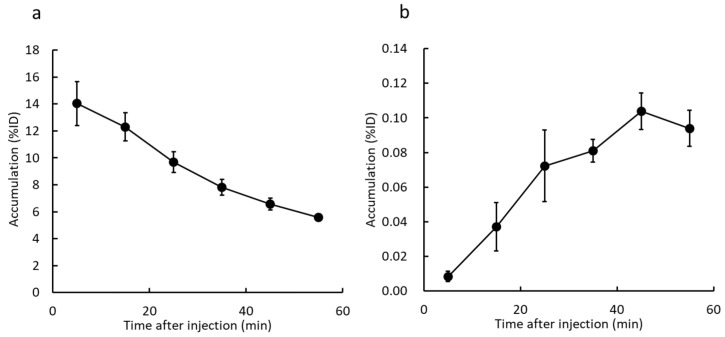
Time activity curves in liver (**a**) and gallbladder (**b**). Mice were scanned from 0 to 60 min after ^123^I-ODV injection. The accumulation in liver decreased, while that in the gallbladder was gradually increased.

**Figure 8 ijms-23-11458-f008:**
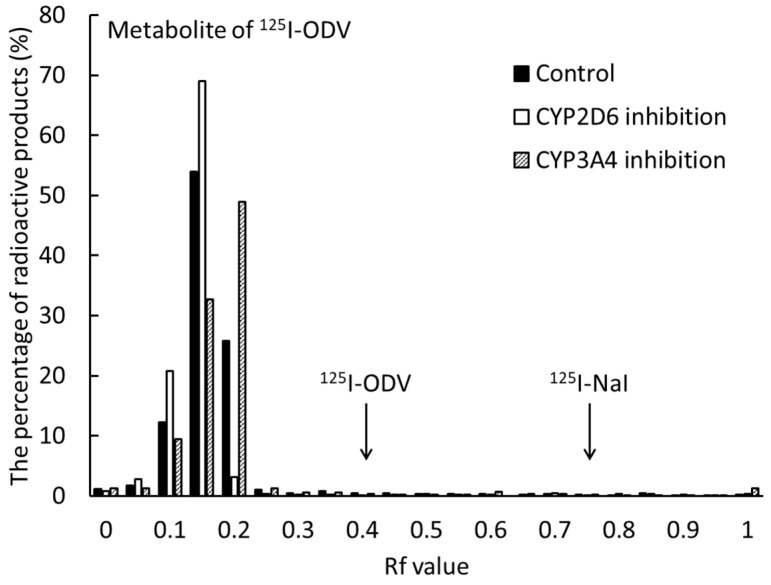
TLC analysis radioactive metabolites in the bile of normal and CYP-inhibited mice after ^125^I-ODV injection. The fractions of ^125^I-NaI and ^125^I-ODV were not found in all conditions, so the metabolite of ^125^I-ODV was selectively excreted in the bile.

**Figure 9 ijms-23-11458-f009:**
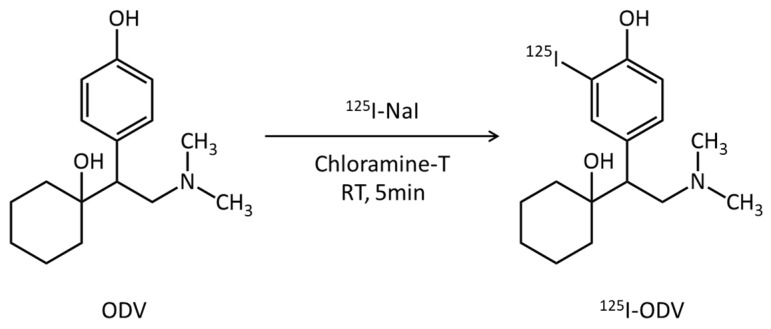
^125^I-labeling scheme of ODV. ODV was labeled with ^125^I-NaI by oxidation reaction.

**Table 1 ijms-23-11458-t001:** Biological distribution of ^125^I-ODV and metabolite of ^125^I-ODV (control).

Organ (%ID/g)	5 min			15 min			30 min			60 min		
Blood	1.89	±	0.03	1.31	±	0.12	1.34	±	0.29	0.74	±	0.22
Brain	0.80	±	0.12	0.75	±	0.05	0.73	±	0.08	0.60	±	0.05
Heart	5.64	±	0.50	2.71	±	0.18	2.04	±	0.12	1.00	±	0.24
Lung	27.04	±	1.59	16.75	±	2.02	13.30	±	2.64	5.01	±	1.14
Thyroid *	0.06	±	0.01	0.03	±	0.02	0.04	±	0.00	0.02	±	0.00
Gallbladder	4.31	±	0.39	5.26	±	1.58	24.23	±	3.53	33.05	±	7.75
Liver	6.91	±	1.05	6.04	±	0.94	5.45	±	1.03	2.89	±	0.61
Spleen	9.22	±	2.16	4.79	±	0.49	3.45	±	0.32	1.75	±	0.41
Pancreas	7.87	±	0.60	6.04	±	0.90	5.02	±	0.73	2.92	±	0.73
Stomach *	2.44	±	0.31	3.19	±	0.42	2.75	±	0.70	1.88	±	0.29
Small intestines *	6.85	±	1.35	7.70	±	0.85	8.47	±	0.87	11.03	±	3.03
Large intestines *	1.32	±	0.16	1.18	±	0.46	0.68	±	0.15	0.47	±	0.05
Kidney	22.95	±	0.43	16.37	±	0.63	14.27	±	1.20	9.42	±	2.23
Urine *	0.63	±	0.89	3.37	±	3.92	3.69	±	2.23	7.31	±	2.80

%ID/g means percent injected dose per gram tissue and mean ± standard deviation obtained from four mice. * %ID/organ was calculated from %ID and measured organ weights.

**Table 2 ijms-23-11458-t002:** Biological distribution of ^125^I-ODV and metabolite of ^125^I-ODV (CYP3A4 inhibition).

Organ (%ID/g)	5 min			15 min			30 min			60 min		
Blood	1.91	±	0.40	1.71	±	0.14 ^‡^	1.44	±	0.15	0.65	±	0.02
Brain	0.71	±	0.03	0.80	±	0.09	0.74	±	0.05	0.52	±	0.03
Heart	5.73	±	0.81	3.93	±	0.37 ^‡^	2.34	±	0.27	0.96	±	0.03
Lung	29.72	±	1.47	21.72	±	3.69	11.31	±	2.60	5.81	±	0.35
Thyroid *	0.05	±	0.01	0.05	±	0.01	0.09	±	0.01 ^‡^	0.11	±	0.01 ^‡^
Gallbladder	2.39	±	1.90	5.43	±	1.63	9.16	±	5.20 ^‡^	20.14	±	1.72 ^‡^
Liver	5.66	±	0.93	4.72	±	1.40	5.03	±	0.47	2.12	±	0.75
Spleen	6.61	±	0.70	6.35	±	0.86 ^†^	4.03	±	0.74	1.94	±	0.46
Pancreas	8.93	±	0.62 ^†^	5.94	±	0.95	23.69	±	37.81	2.77	±	0.02
Stomach *	2.37	±	0.32	2.34	±	0.53	2.24	±	0.34	1.90	±	0.37
Small intestines *	6.06	±	0.29	6.37	±	1.47	7.71	±	3.01	10.07	±	3.63
Large intestines *	1.19	±	0.12	1.00	±	0.30	1.00	±	0.25	0.68	±	0.02 ^‡^
Kidney	21.72	±	0.85	19.74	±	2.36	15.65	±	1.34	9.60	±	3.05
Urine *	2.10	±	0.37	5.25	±	2.28	5.63	±	2.03	2.01	±	2.27

%ID/g means percent injected dose per gram tissue and mean ± standard deviation obtained from four mice. * %ID/organ was calculated from %ID and measured organ weights. ‡ *p* < 0.01 and † *p* < 0.05 compared with control.

**Table 3 ijms-23-11458-t003:** Biological distribution of ^125^I-ODV and metabolite of ^125^I-ODV (CYP2D6 inhibition).

Organ (%ID/g)	5 min			15 min			30 min			60 min		
Blood	1.83	±	0.47	1.59	±	0.33	1.12	±	0.03	0.67	±	0.30
Brain	0.79	±	0.04	0.82	±	0.18	0.71	±	0.11	0.58	±	0.18
Heart	5.19	±	0.74	2.87	±	0.42	1.47	±	0.08 ^‡^	0.62	±	0.25
Lung	22.13	±	2.64	11.96	±	0.51 ^†^	7.77	±	2.29 ^†^	3.15	±	0.82
Thyroid *	0.05	±	0.01	0.04	±	0.01	0.05	±	0.04	0.10	±	0.09
Gallbladder	1.59	±	1.08 ^‡^	10.66	±	1.56 ^‡^	21.66	±	2.30	26.70	±	10.19
Liver	7.19	±	1.14	5.56	±	1.20	5.08	±	1.32	1.75	±	0.39 ^†^
Spleen	4.62	±	1.25 ^†^	4.97	±	0.92	2.88	±	0.59	1.46	±	0.41
Pancreas	6.02	±	1.68	5.71	±	1.20	3.94	±	0.83	2.51	±	0.65
Stomach *	1.77	±	0.54	2.40	±	0.38	3.20	±	0.70	2.20	±	0.81
Small intestines *	7.24	±	1.56	5.64	±	0.45	10.30	±	1.59	10.85	±	2.01
Large intestines *	1.12	±	0.16	0.93	±	0.06	0.74	±	0.25	0.67	±	0.23
Kidney	19.94	±	1.82	17.13	±	3.30	15.90	±	1.45	8.33	±	2.33
Urine *	0.33	±	0.11	6.18	±	0.48	22.96	±	2.22	33.92	±	21.40

%ID/g means percent injected dose per gram tissue and mean ± standard deviation obtained from four mice. * %ID/organ was calculated from %ID and measured organ weights. ‡ *p* < 0.01 and † *p* < 0.05 compared with control.

## Data Availability

The authors confirm that the data supporting the findings of this study are available within the article.

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
