# Peer review of "Measurement of Hepatic CYP3A4 and 2D6 Activity Using Radioiodine-Labeled *O*-Desmethylvenlafaxine"

_ijms, 2022, doi:10.3390/ijms231911458_

Round 1

Reviewer 1 Report

The manuscript describes the synthesis of 123I- and 125I-labeled venlafaxine derivatives and their respective application as probe substrates for measuring cytochrome p450 dependent drug metabolism by imaging and biochemical analysis in mice.

The manuscript has substantial deficiencies and is not suited for publication.

Authors claim throughout the manuscript that they are studying CYP3A4 and -2D6 dependent metabolism of an ODV derivative in mice. However, mice do not express CYP3A4 or -2D6. The mouse CYP2D orthologues are Cyp2d9, -10, -11, -12, and -13; the mouse CYP3A orthologues are Cyp3a11, -13, and -16 (note: the standardized cytochrome p450 nomenclature uses lower case for the mouse genes and enzymes and authors did not adhere to this convention). At best the authors studied one or more of the mouse CYP2D and -3A orthologues of these two human enzymes.

Figure 1 illustrates 125I-labeling of ODV at a specific carbon atom, but no empirical evidence or literature reference is provided to confirm the indicated labeling position. If the authors want to claim they know the label site, they need to provide credible evidence.

Figure 2 labels claim to identify three peaks as 125I-NaI, 125I-ODV, and 125I-ODV byproduct. But no rationale, either a reference or empirical data, is provided for these identifications. This omission impacts the interpretation of each graph that claims to track 125I-ODV (e.g., Figures 3, 4, 5, and 6). Authors need to explain how they “confirmed” or “identified” the peaks as stated in lines 208 and 224.

Authors refer to the CYP inhibitors they employed as “specific” for respective Cyp enzymes (lines 148, 151). While those inhibitors are “selective” for the respective human CYP enzymes, calling them “specific” is an overstatement. This should be changed to “selective” and references should be provided that demonstrate this selectivity is conserved for the mouse orthologues.

The amounts of 123 or 125I-ODV injected into mice is described in terms of specific activity (MBq or kBq in lines 165, 182, 189, and 271). This does not reveal what concentration of the drug was injected, so we don’t know if the mice received a pharmacologically relevant dose and if the subsequent image and organ analysis is physiologically relevant. While specific activity values are useful, the authors need to clarify what concentrations of the radiolabeled drug were administered (e.g., distribution after overdose or underdose may be misleading).

Figure 5 is not sufficiently labeled. Open circles presumably represent 125I-ODV and closed circles represent the so-called “unknown radioactive metabolite”. This needs to be indicated by labels or a description in the figure legend.

In describing Figure 5, authors say the 125I-ODV metabolite increased from 15% up to 46% in the presence of NADPH. But the graph shows an increase from 0% up to about 46%. The 15-46% statement needs to be revised to reflect what the graph shows.

For tables 1, 2, and 3, and Figure 8, authors claim they are tracking over a 1 hour time frame, the tissue distribution and redistribution of injected 125I-ODV. However, they have already shown that about half of the presumed 125I-ODV is converted by liver homogenates to an unknown 125I-labeled metabolite over the same time frame. Minimally they should change table descriptions and labels to indicate that they are tracking 125I-ODV and its metabolites.

The authors interpretation of Fig 9 in light of the figure 7 image analysis are hard to follow. They claim the images show 125I-ODV is excreted in bile (line 368) but then argue from Figure 9 that there is no 125I-ODV in bile. When they say 125I-ODV is excreted in bile (line 368), if they mean the 125I-ODV metabolite, this should be stated clearly.

Figure 9 is hard to interpret as being consistent with the authors assertion that the ODV metabolite, not the parent compound, is excreted to bile, and that metabolism is CYP-dependent. It appears that CYP inhibition led to greater or lesser metabolite accumulation in bile compared to control depending on the Rf value. This may be due to the way they cut the TLC plate up into 21 fractions (line 116). Perhaps a plot of total metabolite counts contained in more than 1 fraction for each condition would show a control value greater than CYP2D6 or -3A4 inhibited values. This would be consistent with their claim that the metabolite, not the parent compound is excreted to bile, and that metabolism is CYP-dependent. From figure 9 it would be hard to conclude that. I suggest they reconfigure Figure 9 to reflect total metabolite counts.

The authors failed to demonstrate, as they claim (line 34 and 35), that 123I-ODV is a tool for quantitative imaging to estimate hepatic Cyp2d and -3a activity. The images do not discriminate between 125I-ODV as the parent compound and its 125I-labeled metabolite. Therefore, on their own, the images reveal nothing about CYP-dependent metabolism of this probe. If they want to claim that comparing control and CYP-inhibitor treated mouse images would reveal differences that correlate to Cyp activity, they will need to demonstrate this.

While the iodinated probes may have specific value in venlafaxine ADME studies, the authors admit that iodination alters ODV metabolism: without the iodine group, it is not a CYP2D6 substrate (line 353 and 354). This suggests that results with this probe do not accurately reflect ODV metabolism. However, as stated at the beginning of this critique, the authors are actually studying mouse Cyp2d enzymes, not the human CYP2D6. So, the paroxitine-sensitive activity they detected, which they erroneously call CYP2D6, may in fact be unaffected by iodination of the substrate.

125I-ODV may have some general value as a p450 probe substratesfor in vitro studies of CYP induction or inhibition. However, since several widely used non-radioactive CYP assay methods are available that mitigate the hazards associated with radioactivity, it seems unlikely that these radiolabeled probes meet a significant unmet need in the research community.

Reviewer 2 Report

In this work the authors present a novel radiopharmaceutical agent for to estimate hepatic CYP3A4 and CYP2D6 activity. Imaging data in normal mice showed that the administered 123I-ODV accumulated early in the liver and was excreted into the gallbladder. In a quite accurate way the authors are demonstrate that depending on the reaction time, the substrate decreased and the metabolite increased, whereas after 30 min of reaction time, the percentage of substrate and its metabolite became mostly constant.

Overall the work is convincing and well written. I only have a few comments/curiosities.

1. The title. Quantification to estimate hepatic CYP3A4 and 2D6 activity using radioiodine-labeled O-desmethylvenlafaxine. It would be more straightforward to rephrase it: "Measurement of hepatic CYP3A4 and 2D6 activity using radioiodine-labeled O-desmethylvenlafaxine"

2. Please provide the scheme of the enzymatic reaction showing the site of modification on the metabolite to have a clear idea of the tranformation. It can easily integrate figure 1.

3. The compound used in this study has a tertiary amine that could be the substrate of hFMO1 and hFMO3 (10.1016/j.bcp.2021.114763, 10.1007/s12210-016-0583-x, 10.1016/B978-0-12-820472-6.00106-7). Please explain why the metabolite is not affected by FMO metabolism.

4. Uncoupling. CYP3A4 Is well known for being highly uncoupled (10.1016/j.bbapap.2017.07.009, 10.3389/fphar.2017.00121). Did the authors measure ROS in association with product formation or have an idea of moles of product / moles of NADPH consumed?

5. Are there other specific radiopharmaceuticals for 3A4 or 2D6 in literature? How do they compare to this study? 
